# Supporting the Conservation and Restoration OpenLab of the Acropolis of Ancient Tiryns through Data Modelling and Exploitation of Digital Media

Efthymia Moraitou [1], Markos Konstantakis [1,*], Angeliki Chrysanthi [1], Yannis Christodoulou [1], George Pavlidis [2], George Alexandridis [3], Konstantinos Kotsopoulos [4], Nikolaos Papastamatiou [5], Alkistis Papadimitriou [6] and George Caridakis [1]

1. Department of Cultural Technology and Communication, University of the Aegean, 811 00 Mytilene, Greece; e.moraitou@aegean.gr (E.M.); a.chrysanthi@aegean.gr (A.C.); yannischris@aegean.gr (Y.C.); gcari@aegean.gr (G.C.)
2. Athena Research Centre, University Campus at Kimmeria Xanthi, 671 00 Xanthi, Greece; gpavlid@athenarc.gr
3. Department of Digital Industry Technologies, National & Kapodistrian University of Athens, 344 00 Psachna, Greece; gealexandri@uoa.gr
4. Department of Computer Engineering and Informatics, University of Patras, 265 04 Patra, Greece; kkotsopoulos@upatras.gr
5. Research Department, SenseWorks Ltd., 454 44 Ioannina, Greece; papastamatiou.nikos@gmail.com
6. Ephorate of Antiquities of Argolida, Ministry of Culture and Sports, 212 00 Nafplio, Greece; azpapadimitriou@culture.gr
* Correspondence: mkonstadakis@aegean.gr

**Abstract:** Open laboratories (OpenLabs) in Cultural Heritage institutions are an effective way to provide visibility into the behind-the-scenes processes and promote documentation data collected and produced by domain specialists. However, presenting these processes without proper explanation or communication with specialists may cause issues in terms of visitors' understanding. To support OpenLabs and disseminate information, digital media and efficient data management can be utilized. The CAnTi (Conservation of Ancient Tiryns) project seeks to design and implement virtual and mixed reality applications that visualize conservation and restoration data, supporting OpenLab operations at the Acropolis of Ancient Tiryns. Semantic Web technologies will be used to model the digital content, facilitating organization and interoperability with external sources in the future. These applications will be part of the OpenLab activities on the site, enhancing visitors' experiences and understanding of current and past conservation and restoration practices.

**Keywords:** OpenLabs; cultural heritage; digital applications; semantic modelling; conservation; restoration data; Business Process Modeling Notation (BPMN); extended reality

## 1. Introduction

Over the last few decades, Cultural Heritage (CH) institutions have been adopting new practices to improve their services and meet the preferences and needs of potential audiences. One such practice is the transformation of conservation and restoration (CnR) laboratories into OpenLabs, which allow visitors to see the various processes that take place "behind the scenes" [1]. However, due to limitations and risks involved in certain workflows when treating CH materials and artefacts, OpenLabs typically present pre-designed experiments and workflows to the public, resulting in closed and regulated environments [2]. Additionally, communication between the audience and expert staff is limited, and interpretative resources explaining what visitors see are not always available.

CnR documentation, which includes textual and visual records, provides significant information that is interesting for specialists, non-expert CH communities, and the public alike [3,4]. This documentation records the structure, preservation state, environment, and

processes related to diagnosis–analysis and CnR of movable or immovable CH, as well as the materials and techniques used for interventions related to preservation and promotion. The Acropolis of Ancient Tiryns is an example of an ongoing process of producing such documentation, which is usually inaccessible to both expert and non-expert communities.

The way OpenLabs currently operate presents challenges to both the openness of the scientific process and the visitor experience, while the rich CnR documentation is not utilised for dissemination. To address these challenges, the CAnTi (Conservation of Ancient Tiryns) project aims to design and implement a digital integrated approach to support the operation of OpenLabs at the Acropolis of Ancient Tiryns. This approach includes (i) the modelling of documentation data using Semantic Web (SW) technologies in order to organise data and and provide syntactic/semantic interoperability of the data for any future integration with external sources and (ii) the development of virtual and mixed reality applications in order to visualise CnR work and provide cultural visitors with access to CnR documentation produced by specialists.

The rest of this paper is organized as follows. Section 2 provides an overview of related work before presenting the project's innovative digital approach. Section 3 presents the development of the CAnTi ontology and its exploitation for CnR documentation data modelling. Section 4 presents the digitization methods that have been applied as well as the results of the digitization process to date. Section 5 presents the different applications that support the CnR OpenLab. Finally, with Section 6, the paper is drawn to a close by discussing lessons learned and future directions.

## 2. Related Work

### 2.1. Digital Content Supporting OpenLabs

In the context of *OpenLabs*, the integration of digital applications is a common practice that varies based on the goals of each project and the level of engagement desired by participating institutions. Often, a digital platform is established to share information about the OpenLab's operations, the experiences visitors can expect, the types of scientific or technical processes that have been or are currently being conducted around a specific theme, and logistical data related to the program, ticketing, and pricing.

Occasionally, the use of advanced digital media, such as custom-designed applications and video presentations, is incorporated. For example, at the Acropolis Restoration Service in Athens, Greece, an educational program introduced an online game called Glauka to engage children with the conservation and restoration (CnR) efforts at Athenian monuments. The game utilizes role-playing, missions, challenges, and rewards to help cultural visitors learn about specific CnR techniques and practice their newly acquired skills [5]. Furthermore, online video "tours" have been employed for ongoing excavations, showcasing critical aspects of the excavation process and expert interpretations on the project website [6].

Finally, digital presentations are featured alongside live demonstrations in OpenLabs, although such instances are relatively infrequent. An example of this is the "Conservators on Exhibition" program at the Benaki Museum, where video presentations and other activities were presented simultaneously with live conservation and restoration processes (https://shorturl.at/nzFY8, accessed on 30 April 2023). This informative presentation serves as a valuable supplement to ongoing research and technical procedures, enhancing the understanding of spectators and visitors. Clearly, there is considerable potential for expanding upon past initiatives and innovating new digital strategies to draw in new audiences through the aid of *OpenLabs*.

### 2.2. Semantic Modelling of CH and CnR Data

In terms of semantic modelling, ample research efforts have been made in the domain of CH to organise and interlink data and to tackle interoperability issues at both the syntactic and semantic levels. CH institutions and organisations have developed several formal ontologies. An ontology is a formal and explicit specification of a shared conceptualization

in order to handle essential aspects of cultural information management, such as retrieval, integration, reuse, and sharing [7–9]. Accordingly, the CH subdomain of CnR exhibits increasing interest in ontologies to handle the highly heterogeneous and often secluded CnR information.

A widely used top-level ontology for the CH domain is the International Committee of Documentation Conceptual Reference Model (CIDOC CRM) of the International Council of Museums (ICOM) [10]. CIDOC CRM provides the basic concepts and relations related to various CH disciplines, and is extended by ten modular models which cover documentation requirements of specific disciplines of the CH domain (FRBRoo, PRESSoo, CRMinf, CRMarchaeo, CRMsci, CRMgeo, CRMdig, CRMba, CRMtex, CRMsoc) (https://www.cidoc-crm.org/collaborations, accessed on 30 April 2023). CIDOC CRM and its official extensions have been widely used for CnR data modelling through the years [11].

Furthermore, a number of ontologies have been developed for the CnR domain in particular, and have been used for specific services regarding CnR data management, most commonly data integration and data searching services. An example is the *Ontology of Paintings and Preservation of Art* (OPPRA), a semantic model that specialises in the CnR of paintings [12]. Another example is the *Monument Damage Ontology* (MDO) [13]. MDO integrates, organises, and processes diverse information related to damage diagnosis and CnR interventions of historical buildings, and will eventually support documentation and monitoring of damage and potential intervention planning and application [14]. Another example is the PARCOURS semantic model, which has been developed for integrating CnR data from different sources to enable the querying of data in a unified way [15]. Finally, the *Conservation Process Model* (CPM) specialises in CnR of historical buildings, and was developed with two objectives: (i) to represent knowledge about the related CnR processes, and (ii) to facilitate integration, mediation, and interchange of heterogeneous CnR data at both the academic and professional levels [16].

Considering the existing practices and models, the current research aims to exploit existing models as well as to extend them in order to fully represent data regarding the CnR of stone construction material, as well as the digitization of CnR processes and the products of digitization.

### 2.3. 3D Digitization of Tangible CH

Digitization of tangible entities entails the transformation of the real world and its features to a virtual world, specifically, the digital world of computing. This virtual world comes with a particular set of rules, benefits, limitations, and opportunities. Being inherently digital, virtual entities are in essence discretised and quantized approximations of the real entities they are based on. Digitised entities may have a potentially unending life in the digital world, which makes them significantly appealing for CH applications. Because tangible heritage assets are of a three-dimensional (3D) nature, their digitization should follow a 3D digital world representation (provided we do not consider the dimension of time for the moment). 3D digitization of CH [17–20] brings new possibilities for presentation, research, knowledge dissemination, conservation, and physical duplication. The domain of CH is challenging for 3D digitization methods due to the wide range of object types, sizes, materials, and complexity. It is apparent that a complete digital documentation of CH should provide an ontologically complete picture of the objects for the future scholars and citizens. 3D digitization has now become common practice in the field of CH research thanks to decades of research and development.

### 2.4. Mixed Reality and Virtual Reality Educational Applications

A novel method was used to depict the sequence of actions taken by users of mixed reality and virtual reality educational applications as well as the process involved in developing the content, which was based on Kotsopoulos etc. 2020 [21]. The processes and functional requirements of the applications were represented and documented using process modelling diagrams (BPMNs). The awareness of the stakeholders with regard

to the functionality of the apps, their features, and how end users interact with them is improved by this kind of visualisation.

Business process modelling (BPM) is expected to play a significant role in the continued development of AI by presenting sequences of action in applications at a higher level. Similar to how a corporation is blind without its management, machine learning in business applications is blind without the guiding function of BPM.

A proposed research model for representational analysis projects that incorporates different stakeholder perspectives than the one in this study was stated by Recker [22]. The model was applied to analyse Business Process Modeling Notation (BPMN), and the study highlights the importance of validation with users and communication with technique developers.

Understandability is a key quality feature of BPMN diagrams, particularly during the requirements phase of software projects; technical and non-technical stakeholders must read, validate, and review complex business processes to guide further software development [23].

Open standards such as BPMN that represent the use of knowledge and semantic data in applications are valuable tools for application design and for understanding the AI and ML models involved in the development of these data. Understanding complex black-box models can highlight the transparency, audibility, and interpretation of these models. The success of explainable AI models in the future will rely on creating novel human–AI interfaces that can facilitate contextual understanding and enable domain experts to pose questions and hypothetical scenarios. In this way, human-designed processes and human-in-the-loop interactions can provide valuable human experience and conceptual knowledge to AI processes, which the current best AI algorithms lack [24].

### 3. Semantic Modelling of CnR Data of Ancient Tyrins

In order to organise the data produced during the conservation of the Acropolis of Ancient Tiryns, as well as to ensure their syntactic/semantic interoperability with external data sources, a conceptual model has been developed in the form of an OWL ontology. The model describes all the necessary information that is produced in the context of conservation, restoration, and documentation of stone building material.

For the development of the ontology, we followed the general guidelines of the METHONTOLOGY [25]. In particular, the development of the ontology was based on an initial conceptualization which was formed during the study and analysis of the data and documents produced during the CnR work performed on the Acropolis of Ancient Tiryns. In particular, the produced documentation material includes visual (photographs, orthophotographs, designs, 3D models etc.) and textual records (notes, reports, presentations). The textual records are either (i) unstructured, including exclusively free text or (ii) semi-structured, including free text fields which present structure (e.g., "material documentation", "damages documentation"). Moreover, a list of terms was collected which are repeatedly present in the semi-structured documentation (e.g., types of materials, methods, damages).

After developing the basic taxonomy of the ontology, an alignment of its concepts and relations with CIDOC CRM (https://www.cidoc-crm.org/sites/default/files/cidoc_crm_v6.2.1-2018April.rdfs, accessed on 30 April 2023) (version 6.2.1) was conducted. Furthermore, the CRMdig (https://cidoc-crm.org/crmdig/sites/default/files/CRMdig_v3.2.1.rdfs, accessed on 30 April 2023) (version 3.2.1) model has been reused, a compatible model with CIDOC CRM which specialises in the representation of methods and products of digital representations of movable/immovable CH.

The reuse of CIDOC-CRM and CRMdig classes was conducted in two ways, depending on whether the concept to be represented constituted a specialisation of or was semantically equivalent with some CIDOC-CRM/CRMdig class (Figure 1). In the first case, a new class was defined as a subclass of some CIDOC CRM/CRMdig class. For instance, the class *ConservationInterventionEquipment* was defined as a subclass of the CIDOC-CRM class

*E22_Man-Made_Object*, and the class *DigitalReconstruction* was defined as a subclass of the CRMdig class *D2_Digitization_Process*. In total, 35 new classes were added. In the second case, the equivalent CIDOC-CRM/CRMdig class has been identified and marked for future CnR data modelling. For instance, the class *E57_Material* was identified as equivalent to the concept *Material*.

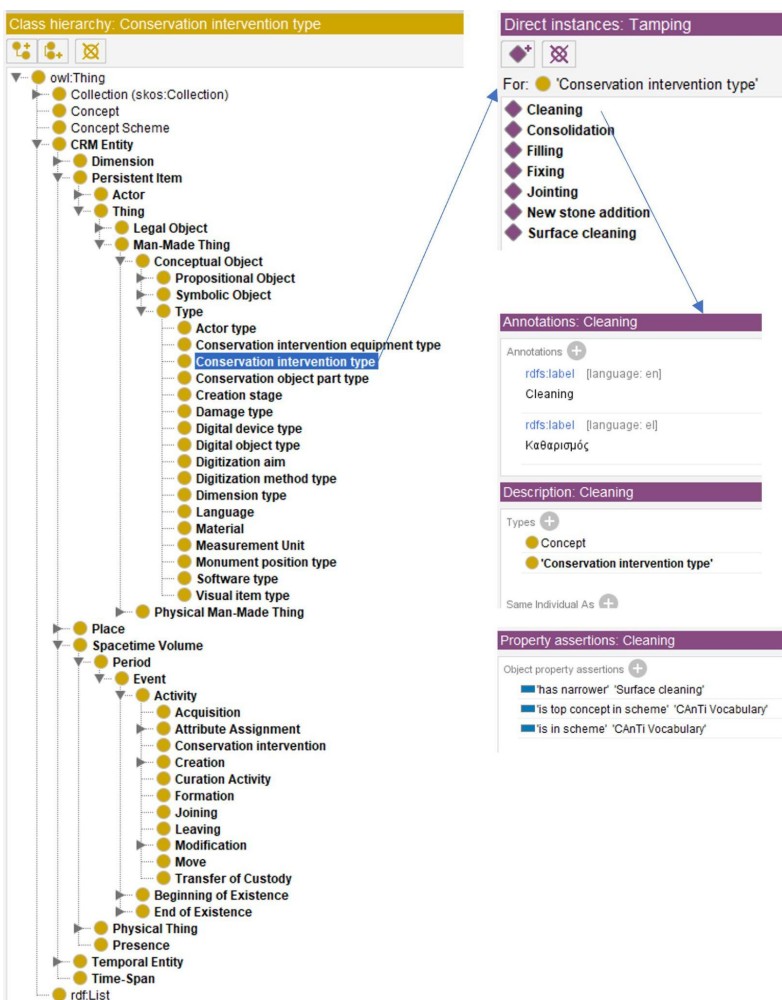

**Figure 1.** Screenshot of a part of the taxonomy of CAnTi ontology and an example of the individuals of interventions' types (using CAnTi ontology and SKOS for their modelling).

Regarding the object/data properties, similarly, either new properties were added or existing properties were identified for future CnR data modelling. For instance, the object property *hasDigitizationAim* (with domain the CRMdig class *D2_Digitization_Process* and range of the added class *DigitizationAim*) was added to the ontology. In total, 18 properties were added. On the other hand, the CIDOC CRM object property *P45_consists_of*, with the domain being the CIDOC CRM class *E18_Physical_Thing* and the range the CIDOC CRM class *E57_Material*, was noted for the CnR data modelling stage.

Furthermore, the SKOS (https://www.w3.org/2004/02/skos/, accessed on 30 April 2023) model was reused for representing the specific types of various basic concepts (e.g., types of materials, types of Acropolis parts, types of interventions) (Figure 1). In order to be usable as an independent thesaurus, these terms were initially organised in a different file and then merged into the ontology. In total, 111 individuals were added, each of which is an individual of (i) the SKOS class Concept and (ii) some sub-class of the CIDOC CRM class *E55_Type*. For instance, the individual for the CnR intervention type cleaning is an individual of the SKOS class *Concept* and the added class *ConservationAndRestorationInterventionType*. In cases where a term is narrower or broader compared to

other terms, the respective individuals are interrelated through the SKOS object properties *has_narrower/has_broader*. For instance, the individuals of the terms *surface cleaning* and *cleaning* are interrelated with narrower/broader relations, as the first term is narrower than the second.

The ontology was developed in Protégé (https://protege.stanford.edu/, accessed on 30 April 2023) (version 5.5.0). The total number of classes, object properties, data properties, and individuals of the ontology, including the extra entities based on the alignment of the ontology with the CIDOC CRM and SKOS ontologies, is presented in Table 1. The terms of the classes, properties, and individuals are included in both Greek and English. The ontology is available at https://github.com/ii-aegean/canti-ontology (accessed on 30 April 2023).

**Table 1.** Number of classes, properties, and individuals of the CAnTi ontology.

| | |
|---|---|
| Class count | 140 |
| Object Property Count | 364 |
| Data Property Count | 27 |
| Individual Count | 111 |

Regarding the modelling of the actual data produced during the CnR work on the Acropolis of Ancient Tiryns, an initial mapping of the work to date was conducted; part of this work is presented in Figure 2. Finally, the ontology served as a guide for building the data schema of the digital repository (see Section 5).

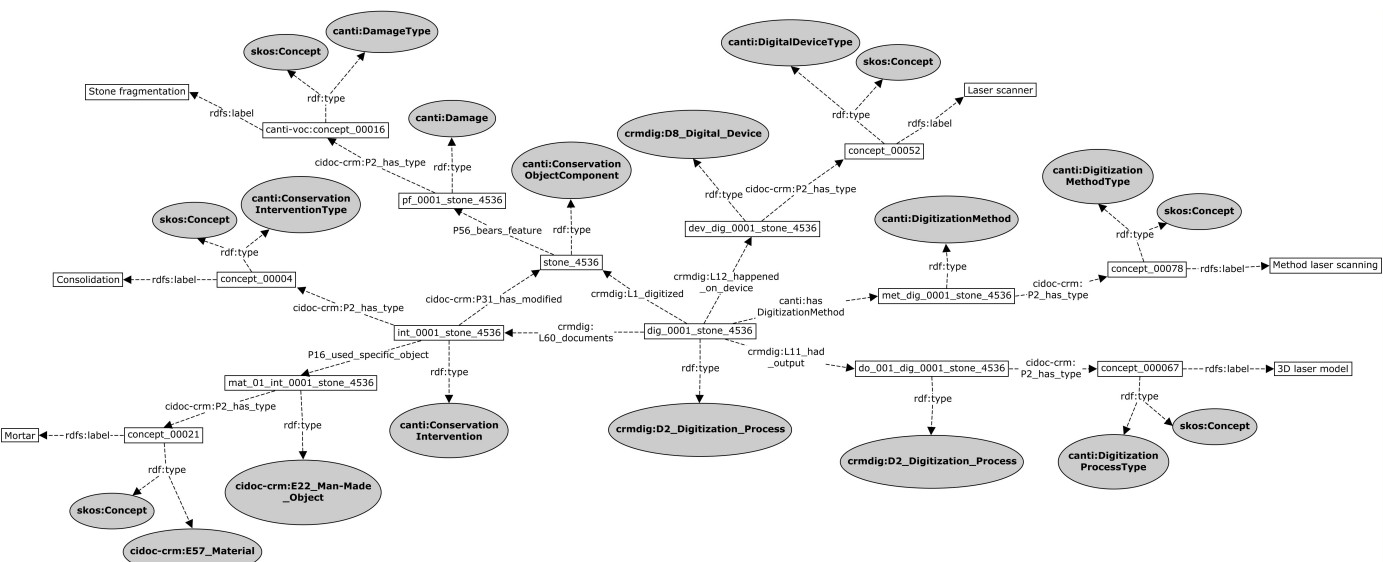

**Figure 2.** An example of modelling of the actual data produced during the CnR work on the Acropolis of Ancient Tiryns using CAnTi ontology.

## 4. Digitization

The site of the Acropolis of Tiryns belongs to the cases of digitization of immovable monuments, where the object is of great size and the control of the parameters and conditions of digitization is complex. As the monument is essentially an architectural complex with an extensive structure, the appropriate method of 3D digitization is a combination of ground and aerial photogrammetry. A representation of the relief of the terrain in the archaeological site is presented in Figure 3. In this representation, the orientation has been preserved.

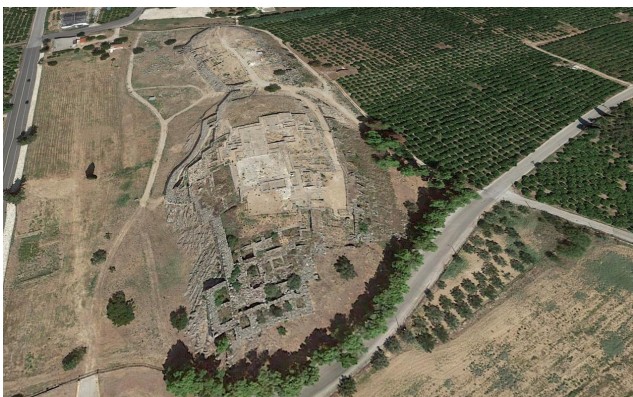

**Figure 3.** 3D representation of Ancient Tiryns from Google Maps.

The maximum North–South and East–West distances (approximately) are estimated to be around 300 m and 100 m, respectively. The perimeter of the monument is estimated to be on the order of 730 m, with an area of approximately 18,000 square metres. It is obvious that this is a monument of large extent and dimensions that poses a challenge for the application of 3D digitization. Among the positive aspects are good accessibility and the absence of objects that could cause visual occlusion (no significant vegetation is present around the monument). Nevertheless, within the monument there are points that present a significant challenge, as there is vegetation and coverage of surfaces. At the initial stages of the project implementation, the necessary data for the 3D reconstruction of the site were collected prior to any restoration intervention of the points of interest. The on-site documentation work led to the collection of around 18,000 photographs (3000 aerial and 15,000 ground-based) with the aim of fully visually documenting the entire monument, with particular emphasis on the area of interest on the southern side. Specifically, 15,000 ground-based photographs were acquired for this southern area, while about two-thirds of the 3000 aerial photographs focused on the area of interest as well.

Figure 4 shows the 3D model of the entire site of Tiryns in the form of a polygon mesh. This particular model was created from 18,643,953 reference points with an RMS reprojection error of 0.284367 (0.714886 pix). The dense point cloud created based on those reference points consisted of 168,482,687 points, while the final 3D polygonal mesh consisted of 33,616,260 faces and 16,816,117 vertices. The texture file of the surface used the UV-Mapping technique and consisted of an image of 4096 $\times$ 4096 pixels. The resolution of the model was estimated as 1.85 cm/pix and the point density as 0.292 points/cm$^2$. In addition, Figure 5 shows the dense 3D point cloud of the region of interest, which will undergo extensive conservation work. This detailed 3D model in point cloud format was created from 5,095,155 reference points with an RMS reprojection error of 0.250799 (3.55413 pix), and the final shown dense point cloud consisted of 1,067,197,447 points.

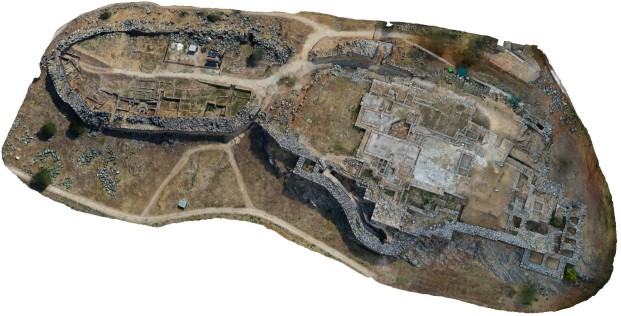

**Figure 4.** 3D reconstruction of the site based on aerial photogrammetry.

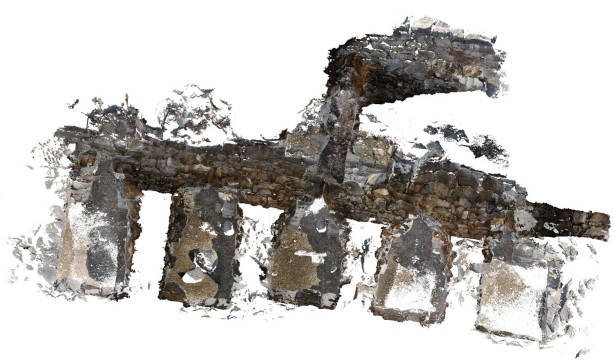

**Figure 5.** Dense 3D point cloud of the region of interest.

## 5. Applications for OpenLab of CnR

This section provides an overview of the applications and services currently under development through the *OpenLab* initiative at the Acropolis of Ancient Tiryns. The primary objective of these efforts is to enhance visitors' cultural experience and promote their familiarity with conservation and restoration (CnR) processes.

Specifically, the development of an online portal has been undertaken, which is designed to facilitate the creation and management of relevant content by CnR specialists. This content includes videos of interventions, documentation comprising text, images, and 3D models, among other things. Concurrently, the general public will have access to the portal both before and after visiting the monument, thereby enabling them to enrich their understanding of the site.

As previously discussed in Sections 1 and 2, the *OpenLab* initiative at the Acropolis of Ancient Tiryns is faced with significant limitations, primarily due to the nature of the interventions required. Among these limitations, one notable challenge is the restricted public access to the intervention locations, with specific provisions in place at the construction site to limit the number of individuals present at any given time. Additionally, the inherent risks associated with the interventions make it unsafe for the audience to be present during the works, while scheduling time for discussion with specialists is a challenging task.

In light of these limitations, it is preferable to design a presentation program that is synchronous with the CnR works, incorporating less dangerous, less complicated, and pre-planned interventions. This approach would enable the safe participation of the audience while ensuring effective communication between specialists and visitors.

The *OpenLab* initiative includes the development of digital applications that aim to enhance the cultural visitors' experience at the Acropolis of Ancient Tiryns. One such application is the Mixed Reality (MR) mobile phone app, which enables visitors to navigate the monument and explore points of interest related to CnR interventions. In addition, this app provides access to visual and textual materials covering the periods before, during, and after interventions, as well as any supplementary documentation.

Another digital application developed through the *OpenLab* initiative is the Virtual Reality (VR) app, which allows cultural visitors to experience a series of interventions in the monument's 3D virtual world. The app enables users to comprehend the different stages of restoration and observe the final aesthetic outcome, all from their mobile or stable device. Visitors can interact with the 3D model to recreate the restoration process.

To further align with the *OpenLab* project's stated objectives, the initiative aims to explore the experimental integration of physical activities with digital applications. Figure 6 below illustrates the nature and specific roles of the *OpenLab* applications and activities.

Overall, these digital applications have the potential to enhance visitors' engagement with the Acropolis of Ancient Tiryns, facilitating a more immersive and informative experience of the monument's history and CnR interventions.

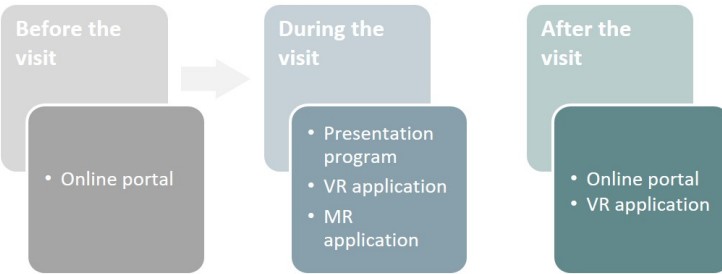

**Figure 6.** OpenLab applications and activities.

Particularly, the requirements of VR and MR applications were represented using the Business Process Model Notation (BPMN). The intention was to make it easier for content creators to perceive how they functioned on a deeper level to allow them to prepare the proper resources as well as for the software development group to realise how they functioned on a more detailed level. These applications have a large number of activity sequences, and the usage of BPMN modelling can assist in more swiftly and efficiently defining the requirements [26].

*5.1. Architecture of the Solution*

For the implementation of the applications, an object-oriented and multi-level approach to the design and organisation of the structures, entities, and individual elements that make up the contents of the application was chosen. It is a four-tier architecture, with the Presentation Layer serving as the top layer for visitors and organisation users. The Service Layer, which offers the required APIs (Web and Internal), comes next, followed by the Business Logic Layer, which contains the application's elements and processes, and finally the Data Layer, which contains the essential data (Figure 7).

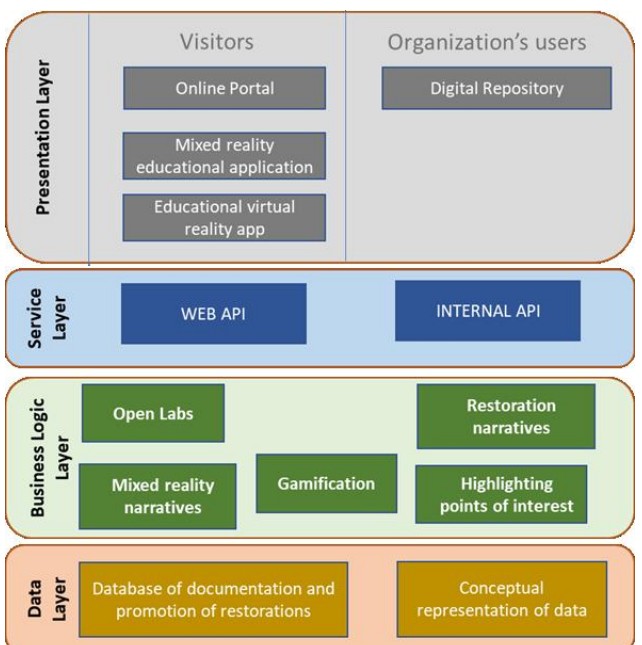

**Figure 7.** Four-Tier Architecture.

The Presentation Layer brings together the requests of the visitors through three interaction environments (online portal, mixed reality educational application, and educational virtual reality app) and the requests of organisation's users through the digital repository, and directs them at the Service Layer by calling the appropriate methods. The answers to the users are returned to the Presentation Layer in descriptive Extensible Markup Language (XML) and JavaScript Object Notation (JSON) format, then the responses are transformed

into a form visible and understandable to the user through graphical user interfaces (GUIs). The Service Layer offers the possibility to expose the business logic layer to different systems through a WEB application programming interface API for applications that need continuous data updating and an INTERNAL API for applications that do not transmit information over the internet. The two APIs are responsible for providing the required information to users. The Business Logic Layer provides the functionality of the services provided, and contains the core services listed below:

- OpenLabs
- Highlighting points of interest
- Mixed-reality narratives
- Gamification
- Restoration narratives

The Data Layer is responsible for providing the necessary data. In order to make it simple to switch to a different database if necessary, the entity classes that interact with the database will be developed using standardised methods of access.

### 5.2. Online Portal

The web portal serves as the primary interface between users and the platform, designed with its target audience and goals in mind. The portal is user-friendly and organised into two areas. The first area is the free access area, accessible to all users without requiring any registration or authentication, which showcases the interventions related to the conservation works at the Acropolis of Tiryns archaeological site. The content is presented attractively based on the chronological order of the works, utilising the logic of OpenLabs. Additionally, it offers informative material about the educational applications of mixed and virtual reality.

The second area is the controlled access area, requiring user authentication and registration. Authorised users can update the web portal's content and add new material. Along with the basic specifications of a modern web portal, the platform supports workflow mechanisms to approve the publication of archaeological material, considering its need for special attention.

It is worth noting that the portal's design considers accessibility and usability for all users, including those with visual or hearing impairments. Its user-friendly interface and content organisation allow for easy navigation, while its controlled access area ensures the accuracy and quality of the published material. The platform's flexibility in supporting different web browsers and connection bandwidths ensures that all users have equal access to its content.

### 5.3. Digital Repository

The digital repository effectively serves two basic operational needs, namely, data aggregation and data search. The organisation of the information managed by the digital repository concerns actions that take place in each archaeological site. These actions may be categorised into:

- Programs: concerns a funding program and any other categorization of interventions under one umbrella.
- Interventions: sets of works concerning a specific part or parts of the archaeological site; an operation may involve excavation, restoration, maintenance or a combination of the above types of intervention.
- Phase: defined work at a specific point of the archaeological site in the context of an intervention.

Each intervention phase is carried out in a specific area or location within the archaeological site, focusing on a particular part, subsection, member, or system of members. Thus, the intervention involves the site itself, one or more intervention points, one or more sections, and one or more subsections within those sections, as well as specific technologies,

machines, means, and materials used in each phase, which are carefully documented for future reference. Additional information such as auxiliary constructions, supplies, and work assignments are registered for ease of access by users.

One of the key aspects of documenting the intervention phases is the use of digital files, which are classified into the categories of visual records and textual records. These digital files are associated with the various stages of the operation, including before, during, and after the intervention, and are accompanied by metadata such as title, description, tags, type, format, and file size. Three-dimensional (3D) files in format, in particular, can be viewed in a specialised interface, enabling users to connect the intervention phases to specific points on the corresponding physical space's 3D model.

Access to the digital repository is granted to all users associated with the archaeological site. The central administrator is responsible for entering information about the site and its specific locations and segments as well as for providing the users with corresponding access privileges to each site.

To summarise, the digital repository application employs a Program/Intervention/ Phase framework to facilitate content registration, and offers documentation tools to each entity by providing lists of options and suggested values for filling in relevant fields. The application includes a file management system capable of handling various formats, and the system is associated with a timestamp and connected to the corresponding intervention phase. As certain files may be too large for server upload, they are instead stored in an external storage space, with the corresponding link stored in the database. Additionally, the digital repository application generates progress reports automatically to gather data from ongoing interventions and their phases for a given project.

### 5.4. Mixed-Reality Educational Application

To improve the representation and documentation of the mixed-reality educational application's processes and functional requirements, a process modelling diagram (BPMN) was created, which is illustrated in Figure 8. A process model is a visual representation of the steps or activities involved in a process. It is a valuable tool for understanding and improving operations, as it allows for identification of how different elements of a process relate to each other, finding bottlenecks or inefficiencies, and making changes to improve overall performance.

In the context of the mixed-reality educational application (MREA), the process model outlines the various steps and activities involved in providing an immersive educational experience for visitors at an archaeological site.

The MREA's process model begins with a start event, where the user activates the application in the area of the monument. The model includes tasks such as the selection of a tour type, searching for points of interest, and selecting a narrative. The model includes exclusive and parallel gateways to control the flow of the process. For example, the exclusive gateway is used to allow users to choose between a free tour and a guided narrative tour. The parallel gateway is used to allow users to search for points of interest while completing other tasks.

If the users choose the self-guided tour option, they can wander freely around the site while being informed about key points of interest and their history through text, photographs, and augmented reality projections. The user can search for points of interest using their mobile phone camera and receive information about each point they visit. After each reading, the user can rate their experience on a scale of 1–5.

If the user chooses a specific narrative, they are presented with a list of available narratives to choose from. Each narrative includes an introductory text that guides the user to the next point of interaction, and the user can participate in mini-games at each interaction point before proceeding to the next one.

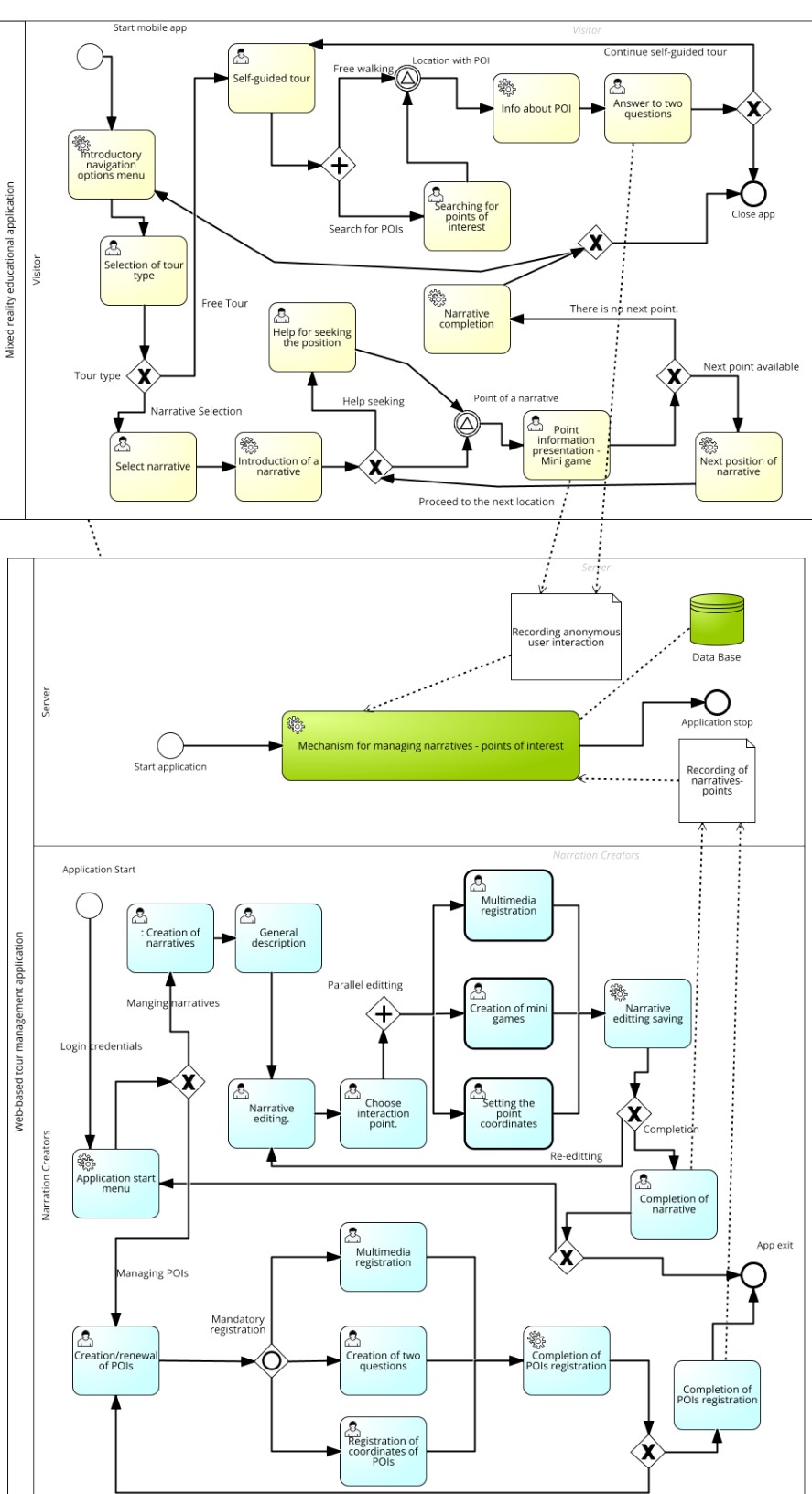

**Figure 8.** The process modelling diagram (BPMN) of the mixed-reality educational application's processes and functional requirements.

The data objects used in the process model are hosted in a database, and include recordings of anonymous user interactions. These recordings can be used to improve the application and provide feedback to users. For example, after visiting a point of interest, the user is asked to rate it and provide feedback on the information presented. These data can be used to improve the accuracy and relevance of the information provided.

The web-based tour management application allows creators to create and manage the content displayed in the mixed reality educational application. Creators can create and edit points of interest, narratives, and general descriptions, as well as manage the content of each narrative. The process model provides a clear and concise way to understand the flow of activities involved in delivering an engaging educational visitor experience at an archaeological site.

### 5.5. Educational Virtual Reality App

In order to better represent and document the processes and functional requirements of the virtual reality education application, a process modelling diagram (BPMN) was created, and is presented in Figure 9.

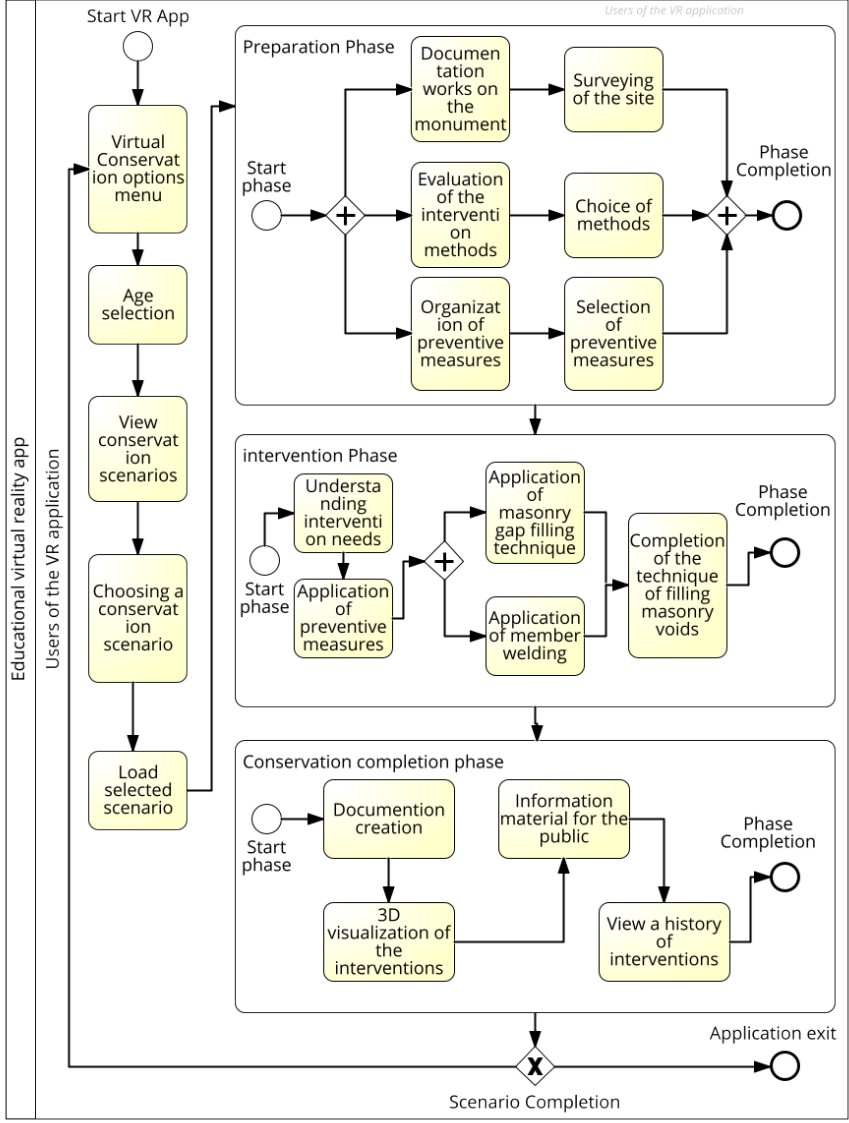

**Figure 9.** The process modelling diagram (BPMN) of the virtual reality educational application's processes and functional requirements.

This representation focuses on the process elements of a virtual reality application designed for conservation interventions at monuments. The process comprises three expanded subprocesses: preparation for conservation, intervention, and conservation completion. The application commences with the entry into the virtual environment, where the user selects their user type from a list that includes adolescents, adults, or specialists. Based on the user's choice, the scenarios and displayed information are tailored to their specific needs.

In the preparation phase, the user is informed about the necessary actions to be taken before the intervention on the monument. This includes learning about monument documentation through surveying the site and analysing its members. The user then selects the most appropriate method of intervention along with the necessary preventive measures and evaluates their chosen method. Multimedia materials are incorporated where necessary to assist the user in understanding the concepts.

The intervention phase involves the application of preventive measures, as well as the use of techniques to fill gaps in the masonry and restore the cohesion of the building. Finally, the user intervenes in a part of the masonry to fill gaps, using intermediate three-dimensional models of the monument through different chronological steps. Multimedia materials such as videos and photos are used where necessary to supplement the intervention process.

The process concludes with the phase completion, where the scenario ends and the user can select another scenario or exit the application. BPMN diagram provides a detailed description of the different tasks and organisation units involved in the educational virtual reality app, supported by text explanations for every task. This study provides valuable insight into the design and implementation of virtual reality applications for conservation interventions at monuments.

## 6. Conclusions

This research work seeks to enhance the operation of the *OpenLab* initiative at the Acropolis of Ancient Tiryns through the design and deployment of a digital integrated approach. The proposed approach leverages software (SW), Mixed Reality (MR), and Virtual Reality (VR) technologies to make CnR documentation material, including textual and visual materials, accessible to the public. In this way, the approach aims to address specific limitations of the *OpenLabs* initiative.

The limitations associated with the case of the Acropolis of Ancient Tiryns include the presentation of complex CnR processes in a non-controllable open-space site and the effective communication of these processes and their outcomes to the public. By utilizing the digital integrated approach, the proposed solution aims to provide visitors with an immersive and informative experience that enhances their understanding of the monument's history and CnR interventions.

Overall, the research work seeks to improve the visitor experience and engagement with the Acropolis of Ancient Tiryns while providing valuable insights into the effectiveness of digital applications in the context of cultural heritage sites.

At this point, the first version of the ontology, which will be used to model CnR data, has been developed. Additionally, the development of the applications which will support the *OpenLab* is in progress. The research work introduces an innovative approach for visualising the complex and dynamic process of creating mixed and virtual reality educational applications. By representing the sequence of actions taken by users and developers, this method can help to enhance the design and functionality of educational apps. The findings have significant implications for the field of education technology, as they provide new insights into the development of effective mixed reality learning experiences.

The ongoing research project involves several subsequent steps, including the completion of the development of the proposed applications and their evaluation in four primary areas. These areas include: (i) the level of public engagement with the CnR domain through the *OpenLabs* initiative, (ii) the extent to which visitors comprehend the presented informa-

tion material regarding CnR interventions, (iii) the user-friendliness and effectiveness of the developed applications, and (iv) the satisfaction of specialists regarding their communication with the public. By evaluating these key areas, the research project aims to provide a comprehensive understanding of the effectiveness of the proposed approach. Specifically, the research seeks to determine the extent to which the digital integrated approach can facilitate the presentation of CnR data by specialists and enhance visitors' engagement and experience. In this way, the project will contribute to the ongoing discourse surrounding the implementation of digital applications in the context of cultural heritage sites.

**Author Contributions:** Conceptualization, E.M. and A.C.; methodology, E.M.; software, K.K. and N.P.; validation, E.M., G.A. and M.K.; formal analysis, E.M. and M.K.; investigation, E.M. and M.K.; resources, E.M. and M.K.; data curation, G.A., E.M. and Y.C.; writing—original draft preparation, M.K., E.M., A.C., Y.C. and G.A.; writing—review and editing, M.K., E.M. and G.A.; visualization, G.P.; supervision, G.C., A.P. and G.P.; project administration, M.K. and E.M.; funding acquisition, M.K. All authors have read and agreed to the published version of the manuscript.

**Funding:** The research of this paper was financially supported by the General Secretariat for Research and Technology (GSRT) through the research project "CAnTi: Conservation of Ancient Tiryns" (T6ΥΒΠ-00271, MIS: 5056234), Framework "Special Actions in aquaculture, industrial materials and open innovation in culture", Operational Program "Competitiveness Entrepreneurship and Innovation", 2014-2021 "Development of Entrepreneurship with Sectoral Priorities", co-financed by the European Regional Development Fund (ERDF) and from National Resources.

**Institutional Review Board Statement:** Not applicable.

**Informed Consent Statement:** Not applicable.

**Data Availability Statement:** Publicly available datasets were analyzed in this study. This data can be found here: https://github.com/ii-aegean/canti-ontology, accessed on 30 April 2023.

**Conflicts of Interest:** The authors declare no conflict of interest.

## Abbreviations

The following abbreviations are used in this manuscript:

| | |
|---|---|
| 3D | Three-Dimensional |
| BPMN | Business Process Model Notation |
| CH | Cultural Heritage |
| CnR | Conservation and Restoration |
| CIDOC CRM | International Committee of Documentation Conceptual Reference Mode |
| CPM | Conservation Process Model |
| DOAJ | Directory of Open-Access Journals |
| GUI | Graphical User Interface |
| JSON | JavaScript Object Notation |
| ICOM | International Council of Museums |
| LD | Linear Dichroism |
| MDO | Monument Damage |
| MDPI | Multidisciplinary Digital Publishing Institute |
| MR | Mixed Reality |
| MREA | Mixed-Reality Educational Application |
| OPRA | Ontology of Paintings and Preservation of Art |
| OWL | Web Ontology Language |
| SM | Semantic Web |
| TLA | Three-Letter Acronym |
| VR | Virtual Reality |

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
