# Peer review of "Supporting the Conservation and Restoration OpenLab of the Acropolis of Ancient Tiryns through Data Modelling and Exploitation of Digital Media"

_computers, doi:10.3390/computers12050096_

Round 1
Reviewer 1 Report
I have carefully read through the paper concerning the“Review Supporting the conservation and restoration OpenLab of the Acropolis of Ancient Tiryns through data modelling and the exploitation of digital media”. The paper is well structured, and the reading is fluent. It is an interesting case study, but I noticed that its novelty and insights are quite low. I have only a small suggestion listed below to improve the study. I recommend the publication after minor revision only if the paper suits very well the aims and purpose of the journal and If case studies are well accepted.
Line 160: please specify what the acronym OWL means.
Figure 3: please add the name of the site. It is obvious, but it is better to specify it.
Line 245: please make 2 apex (cm2).
Figure 6: what do you mean by AR?
Line 318: the sentence seems interrupted.
Line 448: I suggest changing “discussion” with “conclusions”.
The manuscript is well written, it is fluent.
Author Response
Dear Editor & Reviewers,
We would like to thank you for your effort in reviewing our manuscript titled "Supporting the conservation and restoration OpenLab of the Acropolis of Ancient Tiryns through data modelling and exploitation of digital media". We appreciate the thorough reviews and constructive suggestions. We did our best to respond to all the comments and revise the text accordingly (revisions are shown in colour in the revised manuscript). In what follows, we provide brief responses to the reviewers’ comments.
Reviewer #1
- Line 160: Please specify what the acronym OWL means.
- The acronym OWL was defined and a link to was provided as a footnote in the first paragraph of Section 3 (lines 101-102).
- Figure 3: Please add the name of the site. It is obvious, but it is better to specify it.
- The name of the site was added to Figure 3.
- Line 245: Please make 2 apexes (cm2).
- The apex has been fixed.
- Figure 6: What do you mean by AR?
- The incorrect term "AR" was replaced with MR (Mixed Reality).
- Line 318: The sentence seems interrupted.
- The sentence has been fixed.
- Line 448: I suggest replacing “discussion” with “conclusions”.
- The section "Discussion" was updated to "Conclusions."
Reviewer 2 Report
The paper presents the CAnTi (Conservation of Ancient Tiryns) research project. According to the Authors, this paper aims to design and implement virtual and mixed-reality interactive applications that will visualize the conservation and restoration data of the Acropolis of Ancient Tiryns. The topic is interesting and the paper well corresponds with the journal's aims and scope.
However, there are shortcomings in this paper. The abstract should clearly state the aim of the paper. For example please modify the sentence in lines 8-10 to highlight the aim of the paper.
In Section 3 the Authors presented the taxonomy and the ontology. Was any procedure used to build the ontology; e.g. MENTHONTOLOGY, Noy & McGuiness, or similar? This part should be provided and described in detail.
There is also a lack of a definition of ontology. If this term is used, it is worth describing it shortly.
In the article, the Authors use the term "research project" - also in the case of the purpose of the article. Is the research project the goal? Maybe it's worth considering a different wording?
The article also lacks a graphical presentation of the general procedure of the CAnTi (Conservation of Ancient Tiryns) research project. A description, even a general description, of the individual steps of activities carried out by the Authors will definitely improve the readability of the article. In Section 5.1 the Authors presented the architecture of the solution in both described and visualized forms, but due to the large amount of information contained in the article, it is worth considering at least a general presentation of the steps.
The Conclusions section is missing. Section Discussion should sum up the conducted work, but the main clues and highlights should be included in the Conclusions section.
The list of references needs to be extended.
Overall, the article looks good, but it needs improvements.
Minor typos:
Line 214: “…is presented in Figure 2 Finally, the ontology…” – dot is missing
Figures 8 and 9 are illegible
Minor editing of English language required.
Author Response
Dear Editor & Reviewers,
We would like to thank you for your effort in reviewing our manuscript titled "Supporting the conservation and restoration OpenLab of the Acropolis of Ancient Tiryns through data modelling and exploitation of digital media". We appreciate the thorough reviews and constructive suggestions. We did our best to respond to all the comments and revise the text accordingly (revisions are shown in colour in the revised manuscript). In what follows, we provide brief responses to the reviewers’ comments.
Reviewer #2
-
The abstract should clearly state the aim of the paper. For example, please modify the sentence in lines 8-10 to highlight the aim of the paper.
-
In the revised version of the manuscript, the aim of the paper is explained in more detail (lines 8-13).
-
In Section 3 the Authors presented the taxonomy and the ontology. Was any procedure used to build the ontology; e.g. MENTHONTOLOGY, Noy & McGuiness, or similar? This part should be provided and described in detail.
-
For the development of the ontology, the general guidelines of the METHONTOLOGY were followed, and it is explicitly stated in the second paragraph of Section 3 (lines 105-106). Furthermore, the two following paragraphs describe the tasks conducted in the context of the development.
-
There is also a lack of a definition of ontology. If this term is used, it is worth describing it shortly.
-
A definition of the ontology is added in the first paragraph of Section 2.2 (lines 81-83).
-
In the article, the Authors use the term "research project" - also in the case of the purpose of the article. Is the research project the goal? Maybe it's worth considering a different wording?
-
The term “research project” changed to “project”.
-
The article also lacks a graphical presentation of the general procedure of the CAnTi (Conservation of Ancient Tiryns) research project. A description, even a general description, of the individual steps of activities carried out by the Authors will definitely improve the readability of the article.
-
CAnTi project description added in the abstract.
-
In Section 5.1 the Authors presented the architecture of the solution in both described and visualised forms, but due to the large amount of information contained in the article, it is worth considering at least a general presentation of the steps.
-
In the revised version of the manuscript, the presented architecture is explained in more detail (lines 299-303).
-
The Conclusions section is missing. Section Discussion should sum up the conducted work, but the main clues and highlights should be included in the Conclusions section.
-
In the revised version of the manuscript, the Conclusions section is explained in more detail and the Discussion section was added in the Conclusions.
-
The list of references needs to be extended.
-
The references suggested by the reviewer have been included in the revised version of the manuscript.
-
Minor typos: Line 214: “…is presented in Figure 2 Finally, the ontology…” – dot is missing.
-
The typos were fixed.
-
Figures 8 and 9 are illegible
-
In the revised version of the manuscript, Figures 8 and 9 were replaced.
Round 2
Reviewer 2 Report
As it was written in the 1st revision, the paper presents the CAnTi (Conservation of Ancient Tiryns) research project. The Authors were asked to modify the paper. The Authors addressed all the reviewer’s comments. Thank you. I am able to accept this paper in its present form.